# Genomic Analysis of Ceftazidime/Avibactam-Resistant GES-Producing Sequence Type 235 *Pseudomonas aeruginosa* Isolates

**DOI:** 10.3390/antibiotics11070871

**Published:** 2022-06-28

**Authors:** Raúl Recio, Jennifer Villa, Sara González-Bodí, Patricia Brañas, María Ángeles Orellana, Mikel Mancheño-Losa, Jaime Lora-Tamayo, Fernando Chaves, Esther Viedma

**Affiliations:** 1Department of Clinical Microbiology, Instituto de Investigación Sanitaria Hospital 12 de Octubre (imas12), Hospital Universitario 12 de Octubre, 28041 Madrid, Spain; jennifer.villa@salud.madrid.org (J.V.); gonbodsar@gmail.com (S.G.-B.); patriciamaria.branas@salud.madrid.org (P.B.); mariaangeles.orellana@salud.madrid.org (M.Á.O.); fchavessan@gmail.com (F.C.); ester.viedma@salud.madrid.org (E.V.); 2Department of Internal Medicine, Instituto de Investigación Sanitaria Hospital 12 de Octubre (imas12), Hospital Universitario 12 de Octubre, 28041 Madrid, Spain; mikel.mancheno@salud.madrid.org (M.M.-L.); jaime.lora-tamayo@salud.madrid.org (J.L.-T.); 3Centro de Investigación Biomédica en Red de Enfermedades Infecciosas (CIBERINFEC), Instituto de Salud Carlos III, 28029 Madrid, Spain

**Keywords:** *Pseudomonas aeruginosa*, GES *β*-lactamases, ceftazidime/avibactam resistance, whole-genome sequencing

## Abstract

The emergence of ceftazidime/avibactam (CZA) resistance among Guiana extended-spectrum *β*-lactamase (GES)-producing *Pseudomonas aeruginosa* isolates has rarely been described. Herein, we analyze the phenotypic and genomic characterization of CZA resistance in different GES-producing *P. aeruginosa* isolates that emerged in our institution. A subset of nine CZA-resistant *P. aeruginosa* isolates was analyzed and compared with thirteen CZA-susceptible isolates by whole-genome sequencing (WGS). All CZA-resistant isolates belonged to the ST235 clone and O11 serotype. A variety of GES enzymes were detected: GES-20 (55.6%, 5/9), GES-5 (22.2%, 2/9), GES-1 (11.1%, 1/9), and GES-7 (11.1%, 1/9). WGS revealed the presence of two mutations within the *bla*_GES-20_ gene comprising two single-nucleotide substitutions, which caused aspartic acid/serine and leucine/premature stop codon amino acid changes at positions 165 (D165S) and 237 (L237X), respectively. No major differences in the mutational resistome (AmpC, OprD porin, and MexAB-OprM efflux pump-encoding genes) were found among CZA-resistant and CZA-susceptible isolates. None of the mutations that have been previously demonstrated to cause CZA resistance were observed. Different mutations within the *bla*_GES-20_ gene were documented in CZA-resistant GES-producing *P. aeruginosa* isolates belonging to the ST235 clone in our institution. Although further analysis should be performed, according to our results, other resistance mechanisms might be involved in CZA resistance.

## 1. Introduction

A great concern for *Pseudomonas aeruginosa* infections is the global emergence of multidrug-resistant (MDR) and extensively drug-resistant (XDR) isolates, which can cause treatment failure and increased mortality [1]. The successful selection of chromosomal mutations and the growing acquisition of transferable resistance determinants, particularly those encoding carbapenemases (e.g., GES, KPC, VIM, and IMP enzymes) or extended spectrum *β*-lactamases (ESBLs), frequently co-transferred with aminoglycoside-modifying enzyme determinants (e.g., AAC (3′), AAC (6′), and ANT (2′)-I) are responsible for this increasing threat [2,3]. Furthermore, some MDR/XDR *P. aeruginosa* strains, denominated high-risk clones, have a clonal epidemic population structure with limited sequence types (e.g., ST235, ST175, ST111, and ST244) and a well-described ability to disseminate and cause severe infections [4,5,6].

The ceftazidime/avibactam (CZA) association, combining a cephalosporin with a novel established *β*-lactamase inhibitor, was recently introduced in clinical practice. Although unaffected by AmpC enzymes, ESBLs, and class-A carbapenemases (such as GES enzymes), CZA was shown to be ineffective against metallo-*β* lactamase (MBL)-producing isolates [7]. Several studies have demonstrated that CZA displays good in vitro activity against carbapenem-resistant *P. aeruginosa* isolates [8,9]. This novel combination has also been used for treating complicated MDR/XDR *P. aeruginosa* infections [10,11]. Nevertheless, other studies have reported limited CZA activity against carbapenemase-producing *P. aeruginosa* isolates [12,13].

In our institution, the production of GES enzymes is one of the most common causes of multidrug resistance in *P. aeruginosa* isolates [14,15]; thus, CZA is a valuable potential alternative against MDR/XDR *P. aeruginosa* infections. However, various GES variants resistant to CZA began to emerge last year. In general, GES enzymes confer resistance to penicillins, including ureidopenicillins, and oxyimino-cephalosporins but show less activity against carbapenems [2]. Nonetheless, specific substitutions can significantly alter this susceptibility profile, including G170S, which improves the hydrolyzing activity against carbapenems [3]. Whole-genome sequencing (WGS) techniques are promising tools for providing sufficient and reliable data for the surveillance and monitoring of antimicrobial resistance [16]. This study aimed to analyze the molecular epidemiology and resistome of a subset of CZA-resistant GES-producing XDR *P. aeruginosa* isolates by WGS, focusing on the mechanisms involved in CZA resistance, as a part of an institutional surveillance study.

## 2. Methods

### 2.1. Bacterial Sample Collection

All non-duplicated XDR *P. aeruginosa* isolates were collected from the Hospital Universitario 12 de Octubre, a 1300-bed tertiary-care hospital in Madrid, Spain, during 2020. The isolates were recorded at the Clinical Microbiology Laboratory from inpatients admitted to medical or surgical wards of the hospital. Only XDR isolates that were confirmed to be CZA-resistant GES-producing *P. aeruginosa* were further characterized. For a precise comparative analysis, a subset of well-characterized GES-producing XDR *P. aeruginosa* isolates with a CZA-susceptible phenotype were selected for WGS analysis [15].

### 2.2. Clinical Data Collection

Patient data were collected via chart review and included the following: age; gender; comorbidities; ward of admission (intensive care, medical, or surgical); sample type (respiratory, urinary, bloodstream, skin and soft tissue, or colonization); antimicrobial treatment received in the previous month; prior known MDR/XDR *P. aeruginosa* colonization; intensive care admission in the previous month; primary reason for hospital admission; antimicrobial therapy; and outcome (hospital discharge or death).

### 2.3. Microbiological Methods

The identification of *P. aeruginosa* isolates was carried out using matrix-assisted laser desorption/ionization time-of-flight mass spectrometry (MALDI-TOF MS) (Bruker Daltonics Inc., Bremen, Germany). Antimicrobial susceptibility testing was performed using a semi-automated microdilution system (MicroScan, Beckman Coulter Diagnostics, Indianapolis, IN, USA), including the following antimicrobial agents: ceftazidime, cefepime, aztreonam, piperacillin/tazobactam, imipenem, meropenem, gentamicin, tobramycin, amikacin, ciprofloxacin, and colistin. Additionally, ceftolozane/tazobactam and CZA minimum inhibitory concentrations (MICs) were determined by gradient strips (bioMérieux, Marcy l’Etoile, France). MIC_50_ and MIC_90_ values were determined, and percentages of susceptible (standard dose and incremented exposure) and resistant isolates were calculated using European Committee on Antimicrobial Susceptibility Testing (EUCAST) v. 10. 0 (2020) clinical breakpoints (www.eucast.org (accessed on 6 March 2021)). Isolates were considered XDR if they were non-susceptible to at least one agent in all but two or fewer antimicrobial categories [17]. For this study, fosfomycin was not considered. Carbapenemase genes (*bla*_GES_) were screened by polymerase chain reaction (PCR) using LightCycler 2.0 (Roche) and Sanger sequencing using an ABI prism 3100 DNA Sequencer (PE Applied Biosystems, Norkwalk, CT, USA). *P. aeruginosa* ATCC 27853 was used as a quality control for antimicrobial susceptibility testing.

### 2.4. Whole-Genome Sequencing (WGS) and Bioinformatics Analysis

WGS was performed in all *P. aeruginosa* isolates, prepared using indexed pair-end Nextera XT library according to manufacturer’s instructions and sequenced on the MiSeq platform (Illumina Inc., San Diego, CA, USA) with the MiSeq reagent kit v. 2 (Illumina Inc.), resulting in 250-bp paired-end reads. The generated raw reads were quality-trimmed with Trimmomatic tool v. 0. 32 [18] and de novo assembled using the SPAdes assembler v.3.11.1 [19]. Genome assembly was evaluated by QUAST v.5.0.2 [20]. Bacterial identification was confirmed using KmerFinder [21]. Assembled sequences were annotated using Prokka v.1.13.3 [22]. Antimicrobial resistance genes were identified using Antimicrobial Resistance Identification by Assembly (ARIBA) v.2.6.1 [23] with the Comprehensive Antibiotic Resistance Database (CARD) [24] and ResFinder [25] databases. A total of 65 chromosomal genes were selected, and the main relevant mutations known to be involved in antimicrobial resistance were investigated (Appendix A) [2,3]. The in silico multilocus sequence type (MLST) was determined using a BLAST-based approach. The PAst program (https://cge.cbs.dtu.dk/services/PAst-1.0 (accessed on 20 April 2021)) was used for in silico O-serotyping [26]. To determine core genome single-nucleotide polymorphisms (SNPs), all genomes were aligned to the *P. aeruginosa* PAO1 reference genome (GenBank accession: NC_002516.2) using Snippy v.4.6.0 (https://github.com/tseemann/snippy (accessed on 24 April 2021)). Additionally, a non-recombinant core genome SNP was performed using ModelFinder and IQ-TREE v.1.6.3 with 1000 bootstrap replicates [27,28] and visualized using the iTOL tool (https://itol.embl.de/ (accessed on 4 May 2021)). Sequence files were deposited in GenBank under BioProject PRJNA697852 and PRJNA723160 and accession numbers JAFBXZ000000000-JAFCAB000000000 and JAGSOK000000000-JAGSOS000000000.

### 2.5. Ethical Consideration

This study was designed and performed in accordance with the ethical standards of the Helsinki Declaration. The study protocol was approved by the Clinical Research Ethics Committee of our institution (Instituto de Investigación Sanitaria Hospital 12 de Octubre imas12, Hospital Universitario 12 de Octubre, ref.: 19/441). The need for written informed consent was waived due to the retrospective and non-interventional study design.

## 3. Results

### 3.1. Bacterial Isolates and Clinical Data

During a one-year survey study, a total of 102 non-duplicated XDR *P. aeruginosa* isolates were recovered. Of them, 32 (31.4%) were GES producers. Nine (28.1%) isolates were CZA-resistant GES-producing *P. aeruginosa* and all were selected for further genomic analysis. This study also included 13 previously well-characterized CZA-susceptible GES-producing XDR *P. aeruginosa* isolates [15]. The demographics and clinical characteristics of the patients with CZA-resistant and CZA-susceptible isolates, respectively, are shown in Table 1. Overall, most patients colonized or infected by CZA-resistant GES-producing XDR *P. aeruginosa* isolates were older adults, with several underlying conditions and previous broad antibiotic exposure, mainly carbapenems and colistin. However, treatment with CZA prior to isolation was only documented in two (9.1%) patients.

### 3.2. Antimicrobial Susceptibility Data

The antimicrobial susceptibility data are displayed in Figure 1. Increased MICs were observed for aztreonam (from 4/> 16 mg/L to 8/> 16 mg/L) and ceftolozane/tazobactam (from 6/24 mg/L to 24/> 256 mg/L) between CZA-resistant and CZA-susceptible isolates. The most active antipseudomonal agent in CZA-resistant isolates was colistin (100%, MIC_50/90_ = 1/2 mg/L). For the CZA-susceptible isolates, the most active antipseudomonal agents were colistin (100%, MIC_50/90_ = 1/2 mg/L) and CZA (100%, MIC_50/90_ = 2/6 mg/L). The activity of all other antibiotics was lower in both groups. All isolates were classified as XDR phenotypes. The carriage of the *bla*_GES_ genes were confirmed in all isolates.

### 3.3. Molecular Epidemiology

Genomic analysis showed that all isolates belonged to the widespread *P. aeruginosa* high-risk clone ST235. A core-genome phylogenetic tree reconstruction of all isolates and the PAO1 reference isolate is shown in Figure 2. According to the CZA phenotype, isolates were categorized as belonging to two different cluster types (CT-1 and CT-2). The isolates belonging to CT-1 included CZA-resistant phenotypes, and CT-2 comprised CZA-susceptible isolates. A core SNP analysis showed that the genetic diversity of the CZA-resistant and CZA-susceptible isolates ranged from 25 to 21 and from 19 to 83 SNPs, respectively (Appendix A). In silico O-antigen serotyping confirmed that all isolates belonged to the O11 serotype.

### 3.4. Acquired Resistome

The most frequent horizontal acquired *β*-lactamases and aminoglycoside-modifying enzymes are summarized in Figure 3. Genomic analysis revealed a wide variety of GES enzymes among the CZA-resistant isolates. Among them, GES-20 (55.6%, 5/9) was the most frequent carbapenemase documented. The BLAST analysis of bla_GES-20_ revealed the presence of two mutations comprising two single-nucleotide substitutions (G to A and T to G), which caused aspartic acid/serine and leucine/premature stop codon amino acid changes at positions 165 (D165S) and 237 (L237X), respectively. GES-5 (22.2%, 2/9), GES-1 (11.1%, 1/9), and GES-7 (11.1%, 1/9) *β*-lactamases/carbapenemases were also detected. Additionally, OXA-2 β-lactamases were found to be mainly associated with GES-20 carbapenemases. Mutations in OXA-2 (D149, OXA-539) related to CZA resistance were not found. In CZA-susceptible isolates, GES-5 (76.9%, 10/13) was the most frequent carbapenemase, distantly followed by GES-1 (15.4%, 2/13) and GES-20 (7.7%, 1/3). A wide range of antimicrobial resistance determinants (aacA4, aac(6′)-33, aadA1, and aadA6) conferring co-resistance to amynoglycosides were also found in both CZA-resistant and CZA-susceptible isolates.

### 3.5. Mutational Resistome

The complete list of chromosomal genes and mutations investigated is shown in Appendix A. Up to 60.0% (39/65) of the analyzed genes showed non-synonymous or missense mutations. Figure 3 shows the main chromosomal genes involved in resistance to β-lactams, aminoglycosides, and fluoroquinolones. No major differences in the mutational resistome were found among CZA-resistant and CZA-susceptible isolates. All isolates contained non-synonymous mutations in the ampC gene (G27D, A97V, T105A, V205L, and G391A). Notably, none of the CZA-resistant isolates showed ampC substitutions (T96I, G183D, and E247G) known to be related to CZA resistance. Additionally, mutations in other well-known AmpC regulator genes, such as ampR and ampD, were also detected. However, previously described mutations (R504C and F533L) in ftsI (PBP3) related to CZA resistance were not found. Genes involved in the expression and regulation of efflux pumps were frequently mutated, including inactivating mutations in the MexAB-OprM-negative regulators mexR/nalB and nalC. Another frequently mutated gene was oprD, including mutations suggestive of OprD deficiency. Other mutations detected among all isolates included quinolone-resistance-determining region (QRDR) mutations gyrA (T83I) and parC (S87L), which are known to cause fluoroquinolone resistance. Finally, five CZA-resistant isolates showed a fusA1 (elongation factor G) mutation (F582I), linked to aminoglycoside resistance.

## 4. Discussion

Following the introduction of the novel CZA combination for the treatment of GES-producing XDR *P. aeruginosa* infections in our institution, the emergence of resistance to this antimicrobial agent was documented in vitro. Exposure to broad-spectrum antibiotics, including CZA, has been described as one of the main factors related to CZA resistance [1]. Despite this, the confirmation of CZA treatment prior to isolation was documented in a few patients in our cohort. In this study, we used a WGS approach to analyze the genomic characteristics of a subset of CZA-resistant GES-producing XDR *P. aeruginosa* isolates collected at a tertiary hospital as part of a surveillance study. We also focused on the main acquired and mutational antibiotic resistance determinants known to be involved in CZA resistance.

Depending on the underlying mechanisms of antimicrobial resistance, CZA could be an appropriate option for some MDR/XDR *P. aeruginosa* isolates, such as those harboring class-A carbapenemases (such as GES enzymes) or chromosomal combinations (such as OprD deficiency and AmpC hyperproduction) [8,9,10,11,12,13]. In a recent Spanish nationwide study, Del Barrio-Tofiño et al. demonstrated that the prevalence of MDR/XDR *P. aeruginosa* isolates due to transferable ESBLs or carbapenemases was 16.7%, with VIM being the most frequent carbapenemase detected (9.5%), distantly followed by GES enzymes (3.6%) [3]. In our institution, up to 59% of the MDR/XDR isolates demonstrated acquired GES or VIM carbapenemases. This prevalence is higher than that reported in other areas of Spain [2,3,4,6]. In 2019, the distribution of each of these two carbapenemases was 50%. However, in the last year, we experienced a dramatic increase in GES enzymes (81%). The cases reported here offer several important insights into the evolving landscape of GES-producing XDR *P. aeruginosa* isolates in a high-endemicity setting for high-risk clones [14,15].

In silico MLST analysis demonstrated that all isolates belonged to the epidemic *P. aeruginosa* high-risk clone ST235. In addition, core SNP-based phylogenetic analysis confirmed the high diversity among these isolates, suggesting that person-to-person transmission is scarce. The available evidence indicates that ST235, the founder of the CC235 clonal complex, is the most relevant *P. aeruginosa* high-risk clone [4,5,6]. It shows a worldwide dissemination and is associated with MDR/XDR phenotypes by the acquisition of different ESBLs (e.g., OXA and CTX-M) and class-A and -B carbapenemases (e.g., GES, KPC, VIM, IMP, and NDM) [4]. Indeed, the association of ST235 with horizontally acquired resistance determinants, including integrons, transposons, and plasmids, is overwhelming [6]. In a recent genomic analysis, Treepong et al. suggest that the specific presence of DrpA, a determinant involved in homologous recombination, likely increases the ability of this high-risk clone to acquire and maintain foreign resistance elements at a higher rate than other *P. aeruginosa* high-risk clones [5].

A variety of *bla*_GES_ genes were observed among these isolates, and *bla*_GES-20_ was the most frequent carbapenemase. In general, the GES enzymes were class-A ESBLs, although certain variants (GES-4, -5, -6, -14, -15, -16, -18, -20, and -24) exhibited carbapenemase activity due to the presence of a single missense mutation at nucleotide position 493 (G493A), which changed glycine to serine at amino acid position 170 (G170S) [2,3]. Recently, GES mutations related to CZA resistance have been reported [29]. In this regard, Fraile-Ribot et al. demonstrated that P162S substitution reverted the carbapenemase phenotype determined by the G170S change of GES-5 (into GES-1), significantly increasing ceftolozane/tazobactam and CZA MICs. Of note, our GES-20-producing isolates exhibited D165S and L237X substitutions, and we hypothesize that both amino acid changes and gene expression may influence CZA resistance phenotypes. The functional impact of SNPs on the protein sequence was predicted using Protein Variation Effect Analyzer (PROVEAN) V.1.1.3 [30]. Although functional validation is necessary for the D165S variant, a PROVEAN analysis predicted that it had a deleterious effect on function (score −4.000). However, L237X had a neutral effect on function (score 0). This is a concern for clinical sites where GES enzymes seem to be an important contributor to MDR/XDR phenotypes, such as our institution [14,15]. Further studies of the differential expression and mechanisms underlying *bla*_GES_-conferred CZA resistance in other clonal backgrounds are needed.

Additional mechanisms of resistance to CZA have been proposed [31,32,33]. In a previous study, specific mutations leading to the modification of the AmpC catalytic center were found to be the first step in developing resistance to CZA [32]. The changes in the Ω-loop conferred resistance to ceftolozane/tazobactam and cross-resistance to CZA. Moreover, the presence of an OXA-2 mutant (OXA-539, harboring the duplication of the key residue D149) contributed to resistance to CZA [33]. Likewise, mutations R504C and F533L in *ftsI* (PBP3), MexAB-OprM overexpression, OprD inactivation, and AmpC hyperproduction are well-known to be involved in CZA resistance [31]. Unfortunately, none of the aforementioned mutations that have been previously demonstrated to cause CZA resistance were observed in our isolates, suggesting the involvement of other resistance mechanisms. The potential contribution of these determinants to CZA resistance will be assessed in future clinical and experimental studies.

Our study has some limitations that should be acknowledged. First, it was a retrospective study with a small sample size due to the number of available cases, and the results should be interpreted with caution. Second, our study reflects the experience of a single medical center, and the results may not be applicable to other locations with a different molecular epidemiology. Third, we used the consensus definitions of MDR/XDR *P. aeruginosa* phenotypes offered by Magiorakos et al. [17]. While this proposal was certainly useful for harmonizing the definitions of *P. aeruginosa* resistance profiles, the following issues remain: (1) the result varies depending on whether CLSI or EUCAST clinical breakpoints are considered, and (2) the comprehensive application of the proposed definitions is limited by the lack of EUCAST or CLSI clinical breakpoints for fosfomycin. We used EUCAST (2020) clinical breakpoints, and therefore fosfomycin was not considered. According to previous national surveillance studies [2,3], the prevalence of XDR phenotypes may be slightly underestimated when fosfomycin is not considered. Finally, we did not evaluate gene expression, which could have explained the CZA-resistant phenotypes observed in these GES-producing *P. aeruginosa* isolates. Additional transcriptomic analyses are needed to achieve a better understanding of the influence of resistance genes on the evolution of CZA resistance.

In conclusion, our study illustrates the potential molecular complexity of CZA resistance that can emerge in ST235 MDR/XDR *P. aeruginosa* isolates carrying different GES variants. Notably, two single amino acid substitutions within the *bla*_GES-20_ gene were found in CZA-resistant isolates. In addition, although further analysis should be performed, our results indicated that other resistance mechanisms might be involved in CZA resistance. This emerging scenario highlights the need to optimize the use of current antimicrobial agents to minimize the emergence of resistance and track the evolution of resistance with novel genome-based approaches, as well as the urgent need for novel treatments against MDR/XDR *P. aeruginosa* infections.

## Figures and Tables

**Figure 1 antibiotics-11-00871-f001:**
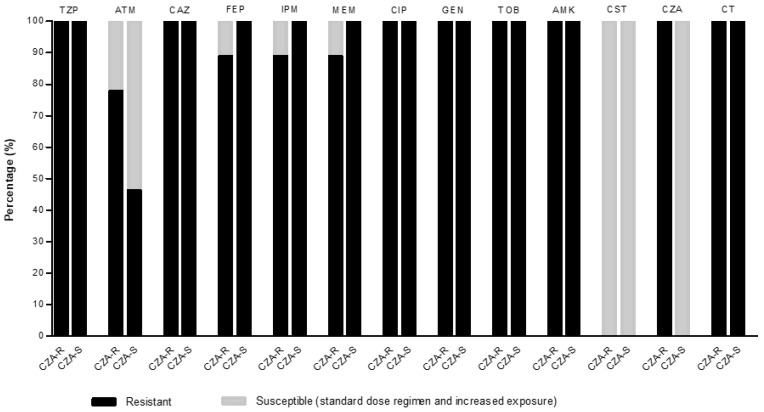
Antimicrobial susceptibility of ceftazidime/avibactam (CZA)-resistant (*n* = 9) and CZA-susceptible (*n* = 13) *P. aeruginosa* isolates. Ceftolozane/tazobactam and CZA minimum inhibitory concentrations (MICs) were determined by gradient strips. Percentages of resistant and susceptible (standard dose and incremented exposure) isolates were calculated using the European Committee on Antimicrobial Susceptibility Testing (EUCAST) v.10. 0 (2020) clinical breakpoints. TZP, piperacillin/tazobactam; ATM, aztreonam; CAZ, ceftazidime; FEP, cefepime; IPM, imipenem; MEM, meropenem; CIP, ciprofloxacin; GEN, gentamicin; TOB, tobramycin; AMK, amikacin; CST, colistin; CZA, ceftazidime/avibactam; CT, ceftolozane/tazobactam.

**Figure 2 antibiotics-11-00871-f002:**
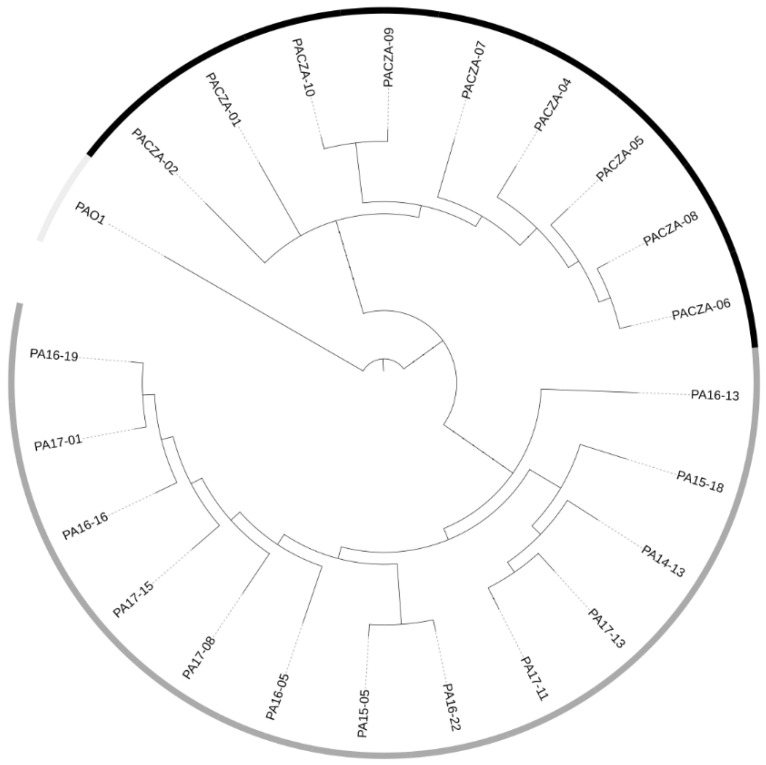
Core-genome maximum-likelihood phylogenetic tree of all *P. aeruginosa* isolates and the *P. aeruginosa* PAO1 reference genome. The units of the scale are single-nucleotide polymorphisms (SNPs) by position. Ceftazidime/avibactam (CZA) phenotypes of *P. aeruginosa* isolates are highlighted by shaded squares: resistant (black), susceptible (dark grey), and not applicable (light grey). CZA, ceftazidime/avibactam.

**Figure 3 antibiotics-11-00871-f003:**
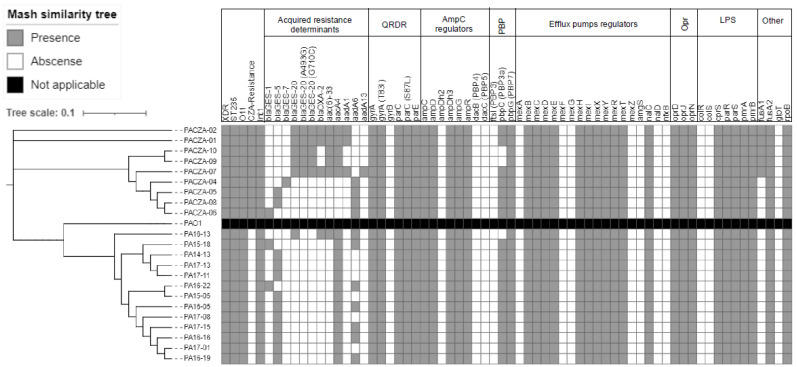
Mash similarity tree of the *P. aeruginosa* isolates analyzed by whole-genome sequencing. Main acquired and mutational genes involved in antimicrobial resistance are also included. Brach length is indicative of the Mash distance. QRDR, quinolone-resistance-determining region; PBP, penicillin-binding protein; LPS, lipopolysaccharide; XDR, extensively drug-resistant; ST, sequence type; CZA, ceftazidime/avibactam.

**Table 1 antibiotics-11-00871-t001:** Demographic and clinical characteristics of the study cohort.

ID	Collection Date	Age	Gender	Ward	Sample Type	Prior Antipseudomonal Antibiotics	Prior CZA	Primary Reason for Admission	Patient Outcome
PACZA-01	2020-01-08	69	Female	ICU	Urine	MEM, CST	None	Hepatic transplant	Death
PACZA-02	2020-08-03	60	Male	Haematology	Respiratory	TZP, MEM, CST	None	Febrile neutropenia	Hospital discharge
PACZA-04	2020-01-09	34	Female	Medical	Blood	MEM	None	Catheter-related bloodstream infection	Hospital discharge
PACZA-05	2020-04-10	64	Female	ICU	Respiratory	TZP, MEM, CST, CZA, AMK	Yes	Ventilator-associated pneumonia	Hospital discharge
PACZA-06	2020-09-04	38	Female	Surgical	Urine	CIP	None	Heart transplant	Death
PACZA-07	2020-09-09	80	Male	Medical	Blood	MEM	None	Urinary tract infection	Hospital discharge
PACZA-08	2020-11-09	81	Male	Medical	Urine	MEM, CST	None	Urinary tract infection	Hospital discharge
PACZA-09	2020-02-11	29	Male	Surgical	Soft tissue	MEM, CZA	Yes	Wound infection	Hospital discharge
PACZA-10	2020-02-14	70	Female	Medical	Colonization	CIP	None	Decompensate heart failure	Death
PA14-13	2014-09-26	82	Male	Medical	Blood	CIP	None	Respiratory tract infection	Death
PA15-05	2015-05-25	68	Male	Medical	Blood	IPM, CIP	None	Decompensation of liver cirrhosis	Death
PA15-18	2015-12-16	57	Male	ICU	Blood	TZP	None	Respiratory tract infection	Death
PA16-05	2016-06-19	59	Male	ICU	Blood	MEM, CST, ATM, AMK	None	Haematopoietic transplantation	Death
PA16-13	2016-09-16	86	Male	ICU	Blood	CIP	None	Schönlein-Henoch purpura vasculitis	Death
PA16-16	2016-10-10	63	Female	Haematology	Blood	MEM, AMK	None	Febrile neutropenia	Death
PA16-19	2016-11-16	51	Male	Haematology	Blood	CIP	None	Febrile neutropenia	Death
PA16-22	2016-02-24	62	Male	Haematology	Blood	MEM, AMK	None	Febrile neutropenia	Death
PA17-01	2017-10-26	76	Male	Oncology	Blood	None	None	Late-stage lung carcinoma	Death
PA17-08	2017-02-11	39	Female	ICU	Blood	None	None	Subdural haematoma	Death
PA17-11	2017-04-02	66	Male	Haematology	Blood	MEM, CIP	None	Leukaemia treatment	Death
PA17-13	2017-05-05	42	Male	ICU	Blood	TZP	None	Coronary acute syndrome	Death
PA17-15	2017-05-18	79	Female	Medical	Blood	TZP, MEM, AMK	None	Paralytic ileus	Death

ICU, intensive care unit; CST, colistin; TZP, piperacillin/tazobactam; ATM, aztreonam; MEM, meropenem; CIP, ciprofloxacin; AMK, amikacin; CZA, ceftazidime/avibactam; IPM, imipenem.

## Data Availability

The data presented in this study are available in the article.

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
