# Peer review of "Genomic Analysis of Ceftazidime/Avibactam-Resistant GES-Producing Sequence Type 235 Pseudomonas aeruginosa Isolates"

_antibiotics, 2022, doi:10.3390/antibiotics11070871_

Round 1

Reviewer 1 Report

The manuscript is well written and organized. The study was properly designed and executed.

The authors investigated the molecular epidemiology and resistome of a subset of CZA-resistant GES-producing XDR P. aeruginosa isolates collected at a tertiary hospital by WGS, focusing on the mechanisms involved in CZA resistance, as a part of a surveillance institutional study.

They were able to identify two mutations within the blaGES-20 gene which caused aspartic acid/serine and leucine/premature stop codon amino acids changes at positions 165 and 237, and hypothesized that both changes and gene expression may influence the CZA resistance phenotype.

The results are also sustained by the expertise of the research group, validated by the referenced papers. Nonetheless, I must refer that those self-citations could be reduced to the essential studies.

On another note, currently undescribed mutations potentially related to resistance could be sorted resorting to in silico prediction tools of the putative functional impact of the amino acid change. Tools like SIFT, PolyPhen-2, or PROVEAN, would predict whether an amino acid substitution is likely to affect protein structure and function based on sequence homology.

Minor comments:

- L. 201 “Adquired” should be corrected to “Acquired”

- In Figure 3, I believe the authors could try to enhance the gene association with each category, making it more visible to the reader

- Table S2 and S3 should indicate the measure of distance for the antibiotic gradient strips.

- Table S3: “suscpetible” should be corrected.

Author Response

POINT-BY-POINT RESPONSE-REVIEWERS´ COMMENTS (antibiotics-1776838)

Reviewer #1

The manuscript is well written and organized. The study was properly designed and executed. The authors investigated the molecular epidemiology and resistome of a subset of CZA-resistant GES-producing XDR P. aeruginosa isolates collected at a tertiary hospital by WGS, focusing on the mechanisms involved in CZA resistance, as a part of a surveillance institutional study. They were able to identify two mutations within the blaGES-20 gene which caused aspartic acid/serine and leucine/premature stop codon amino acids changes at positions 165 and 237, and hypothesized that both changes and gene expression may influence the CZA resistance phenotype.

The results are also sustained by the expertise of the research group, validated by the referenced papers. Nonetheless, I must refer that those self-citations could be reduced to the essential studies.

We thank the reviewer for this suggestion. Accordingly, we have reduced references to essential studies from our research group (please see references #14 and #15, revised version).

On another note, currently undescribed mutations potentially related to resistance could be sorted resorting to in silico prediction tools of the putative functional impact of the amino acid change. Tools like SIFT, PolyPhen-2, or PROVEAN, would predict whether an amino acid substitution is likely to affect protein structure and function based on sequence homology.

We thank the reviewer for this interesting suggestion, since it has helped us to improve our work. The functional impact of single nucleotide polymorphisms (SNPs) D165S and L237X on the protein sequence GES-20 was predicted using Protein Variation Effect Analyzer (PROVEAN) v.1.1.3. Although functional validation is necessary for D165S variant, a PROVEAN analysis predicted to have a deleterious effect on function (score: -4.000). L237X not have deleterious effect on function (score: 0, neutral). In our opinion, a future comparative analysis of the differential expression of GES-20 should be further explored. Accordingly, we have added some additional thoughts in the discussion section and we have added a new reference (#30) (please see lines 290-296 and reference #30, revised version).

  1. 201 “Adquired” should be corrected to “Acquired”

Following the reviewer suggestion, we have change “Adquired” to “Acquired” in this sentence (please see line 204, revised version).

In Figure 3, I believe the authors could try to enhance the gene association with each category, making it more visible to the reader

We have followed the reviewer´s suggestion and have improve the Figure 3 in order to show more visible to the readers. Figure 3 shows a Mash similarity tree of the P. aeruginosa isolates analyzed by whole-genome sequencing. Main acquired and mutational genes involved in antimicrobial resistance are also included. Brach length is indicative of the Mash distance (please see Figure 3, revised version).

Table S2 and S3 should indicate the measure of distance for the antibiotic gradient strips.

We fear that we may have failed to express the actual aim of these supplementary tables, which really was to depict single nucleotide polymorphisms (SNPs) distances between CZA-resistant (Table S2) and CZA-susceptible (Table S3) P. aeruginosa isolates. As you may see in Table S2 and Table S3 SNPs analysis showed that genetic diversity of CZA-resistant and CZA-susceptible isolates ranged from 25-221 and 19-83 SNPs, respectively. Out to clarity, we have improved the titles of supplementary tables (please see Table S2 and Table S3, revised version).

Table S3: “suscpetible” should be corrected.

Following the reviewer suggestion, we have change “suscpetible” to “susceptible” in the Table S3 (please see Table S3, revised version).

Reviewer 2 Report

The title of the article is “Genomic Analysis of Ceftazidime/Avibactam Resistant GES-producing Sequence Type 235 Pseudomonas aeruginosa isolates”. Authors aimed to analyze the molecular epidemiology and resistome of ceftazidime/avibactam (CZA)-resistant GES-producing XDR P. aeruginosa isolates by whole-genome sequencing (WGS), focusing on the mechanisms involved in CZA resistance. I think, the study will be helpful for future study. The manuscript is well written. However, few observations are in bellow to clarify-

The authors may write few sentences on the function of GES and related enzyme on antibiotics and resistance in introduction section. 

Authors may write few more on microbiological methods.

Line 144-145: Authors have mentioned all the isolates as XDR. They need to check this statement. According to the definition of XDR maximum two classes or categories of antibiotic will be susceptible. From the supplementary data, it is clear that few isolates (yellow marked-bellow in table) are susceptible to Aztreonam (Monobactams) and Colistin (Polymyxins). Authors have tested all categories of antibiotics except Phosphonic acids class of antibiotic (Fosfomycin) to find out the XDR nature. Therefore, the status of Phosphonic acids is unknown. In this situation, it is difficult to mention the yellow marked isolates as XDR. Please see the original XDR definition paper (doi: 10.1111/j.1469-0691.2011.03570.x.) for XDR criteria of P. aeruginosa.

Line 15: Please write the full meaning of GES.

Line 93: Please use the original reference for XDR definition. doi: 10.1111/j.1469-0691.2011.03570.x.

Line 130: Did the authors recheck the susceptibility of CZA-susceptible isolates or use previous data?

Isolates

PCZA-01

PCZA-07

PA15-05

PA16-22

PCZA-02

PCZA-08

PA15-18

PA17-01

PCZA-04

PCZA-09

PA16-05

PA17-08

PCZA-05

PCZA-10

PA16-13

PA17-11

PCZA-06

PA14-13

PA16-16

PA17-13

Author Response

POINT-BY-POINT RESPONSE-REVIEWERS´ COMMENTS (antibiotics-1776838)

Reviewer #2

The title of the article is “Genomic Analysis of Ceftazidime/Avibactam Resistant GES-producing Sequence Type 235 Pseudomonas aeruginosa isolates”. Authors aimed to analyze the molecular epidemiology and resistome of ceftazidime/avibactam (CZA)-resistant GES-producing XDR P. aeruginosa isolates by whole-genome sequencing (WGS), focusing on the mechanisms involved in CZA resistance. I think, the study will be helpful for future study. The manuscript is well written. However, few observations are in bellow to clarify.

The authors may write few sentences on the function of GES and related enzyme on antibiotics and resistance in introduction section. 

We thank the reviewer for inviting us to deep in such an interesting aspect. In general, GES enzymes confer antimicrobial resistance to most penicillins, including ureidopenicillins, and oxyimino-cephalosporins, but they show less activity against carbapenems. Nonetheless, specific substitutions can significantly alter this susceptibility profile, including G170S, which improves activity against carbapenems. In order to make it clearer, we have added this information in the introduction section (please see lines 58-62, revised version).

Authors may write few more on microbiological methods.

We have followed the reviewer´s suggestion and have added some additional data in methods section (please see lines 96-99, revised version).

Line 144-145: Authors have mentioned all the isolates as XDR. They need to check this statement. According to the definition of XDR maximum two classes or categories of antibiotic will be susceptible. From the supplementary data, it is clear that few isolates (yellow marked-bellow in table) are susceptible to Aztreonam (Monobactams) and Colistin (Polymyxins). Authors have tested all categories of antibiotics except Phosphonic acids class of antibiotic (Fosfomycin) to find out the XDR nature. Therefore, the status of Phosphonic acids is unknown. In this situation, it is difficult to mention the yellow marked isolates as XDR. Please see the original XDR definition paper (doi: 10.1111/j.1469-0691.2011.03570.x.) for XDR criteria of P. aeruginosa.

We thank the reviewer for this opportunity to clarify the text. For this study, we have used the consensus definitions of MDR/XDR P. aeruginosa profiles by Magiorakos et al. (Clin Microbiol Infect 2012, 18 (3): 268-81) published in 2012. While this proposal was certainly useful for harmonization of definitions of P. aeruginosa resistance profiles, several aspects remain to be considered. First, even if a single definition is used, the result will vary depending on whether CLSI or EUCAST clinical breakpoints are used. Second, the comprehensive application of the proposed definitions is limited by the lack of clinical breakpoints (both CLSI and EUCAST) for fosfomycin. We have used EUCAST (2020) clinical breakpoints and therefore, fosfomycin was not considered. According to previous national surveillance studies (Del Barrio-Tofiño et al. 2017, Del Barrio-Tofiño et al. 2019), the prevalence of XDR phenotypes may be slightly underestimated with respect to the study by Magiorakos et al. since fosfomycin was not considered. This stands as an inherent limitation of the present study, which we have acknowledge in the revised manuscript (please see lines 314-316, revised version).

Line 15: Please write the full meaning of GES.

Following the reviewer suggestion, we have written the full meaning of GES “Guiana extended spectrum b-lactamase” (please see lines 15-16, revised version).

Line 93: Please use the original reference for XDR definition. doi: 10.1111/j.1469-0691.2011.03570.x.

We agree with the reviewer that consensus XDR definitions by Magiorakos et al. (Clin Microbiol Infect 2012, 18 (3): 268-81) are used in this study, as we did acknowledge in the original text. Isolates were considered XDR if they were non-susceptible to at least one agent in all but two or fewer antimicrobial categories. For this study, fosfomycin (lack of clinical breakpoints in EUCAST 2020) was not considered (please see lines 96-97 and reference #17, revised version).

Line 130: Did the authors recheck the susceptibility of CZA-susceptible isolates or use previous data?

We thank the reviewer for this suggestion. The present work follows the lead of our previous study (please see Recio et al. Clinical and Bacterial Characteristics of Pseudomonas aeruginosa Affecting the Outcome of Patients with Bacteraemic Pneumonia. Int. J. Antimicrob. Agents. 2021, 58, 106450), where we observed that CZA might be a good in vitro option for MDR/XDR GES-producing P. aeruginosa isolates. For this reason, we no have retested the susceptibility to CZA. In this regard, the selection of CZA-susceptible isolates has been based in previous published data, but it has also aimed to be representative of the whole cohort.

Isolates

PCZA-01

PCZA-07

PA15-05

PA16-22

PCZA-02

PCZA-08

PA15-18

PA17-01

PCZA-04

PCZA-09

PA16-05

PA17-08

PCZA-05

PCZA-10

PA16-13

PA17-11

PCZA-06

PA14-13

PA16-16

PA17-13

Round 2

Reviewer 2 Report

Just one suggestion, they should add why fosfomycin was not considerded in this study and how XDR status was formulated without fosfomysin by current antibiogran findings.  Authors may add  their response (For this study, we have used the consensus definitions of MDR/XDR P. aeruginosa profiles by Magiorakos et al. (Clin Microbiol Infect 2012, 18 (3): 268-81) published in 2012. While this proposal was certainly useful for harmonization of definitions of P. aeruginosa resistance profiles, several aspects remain to be considered. First, even if a single definition is used, the result will vary depending on whether CLSI or EUCAST clinical breakpoints are used. Second, the comprehensive application of the proposed definitions is limited by the lack of clinical breakpoints (both CLSI and EUCAST) for fosfomycin. We have used EUCAST (2020) clinical breakpoints and therefore, fosfomycin was not considered. According to previous national surveillance studies (Del Barrio-Tofiño et al. 2017, Del Barrio-Tofiño et al. 2019), the prevalence of XDR phenotypes may be slightly underestimated with respect to the study by Magiorakos et al. since fosfomycin was not considered. This stands as an inherent limitation of the present study) in the discussion for the reader understanding. 

Author Response

We thank the reviewer for this opportunity to clarify the text. For this study, we have used the consensus definitions of MDR/XDR P. aeruginosa profiles by Magiorakos et al. (Clin Microbiol Infect 2012, 18 (3): 268-81) published in 2012. While this proposal was certainly useful for harmonization of definitions of P. aeruginosa resistance profiles, several aspects remain to be considered. First, even if a single definition is used, the result will vary depending on whether CLSI or EUCAST clinical breakpoints are used. Second, the comprehensive application of the proposed definitions is limited by the lack of clinical breakpoints (both CLSI and EUCAST) for fosfomycin. We have used EUCAST (2020) clinical breakpoints and therefore, fosfomycin was not considered. According to previous national surveillance studies (Del Barrio-Tofiño et al. 2017, Del Barrio-Tofiño et al. 2019), the prevalence of XDR phenotypes may be slightly underestimated with respect to the study by Magiorakos et al. since fosfomycin was not considered. This stands as an inherent limitation of the present study, which we have acknowledge in the revised manuscript (please revised version).